# Sleep and Perceived Stress: An Exploratory Mediation Analysis of the Role of Self-Control and Resilience among University Students

**DOI:** 10.3390/ijerph20166560

**Published:** 2023-08-11

**Authors:** Silvia Aracely Tafoya, Vania Aldrete-Cortez, Fabiola Tafoya-Ramos, Claudia Fouilloux-Morales, Claudia Díaz-Olavarrieta

**Affiliations:** 1Department of Psychiatry and Mental Health, Faculty of Medicine, National Autonomous University of Mexico, Mexico City 04510, Mexico; stafoya@unam.mx (S.A.T.); cfouilloux@facmed.unam.mx (C.F.-M.); 2Laboratory of Neuroscience and Cognitive Development, Psychology Department, Universidad Panamericana, Mexico City 03920, Mexico; valdrete@up.edu.mx; 3Psychiatric Attention Services, Secretaría de Salud, Mexico City 10200, Mexico; fabiols.tafoya@salud.gob.mx

**Keywords:** stress, sleep quality, self-control, resilience, college

## Abstract

Background: High levels of stress are frequent in university education, and a lack of sleep has been reported to make students more vulnerable to stress. The mechanisms through which sleep harms students have not been sufficiently clarified; therefore, this study aimed to explore the mediating role of self-control and resilience in the relationship between sleep quality and duration and perceived stress. Methods: Of 32 first-year college students, 21 (78%) were women, with a mean age of 18.47 (±0.84). They responded to a self-administered survey that included questions on stress, resilience, and sleep quality and recorded their daily sleep duration using a wristband for six days. Results: Perceived stress was significantly correlated with resilience (r = −0.63), self-control (r = −0.46), sleep duration (r = −0.35), and lower sleep quality (r = 0.57). Path analysis revealed that self-control and resilience were partially mediated by sleep quality (R^2^ = 0.62; *p* < 0.01) and completely mediated by sleep duration (R^2^ = 0.46; *p* < 0.01). In both models, self-control had a direct effect on resilience and had a good-fit index. Conclusion: Being resilient seems to play a mediating role in the relationship between sleep and perceived stress; this ability can be favored by self-control, which is directly influenced by sleep.

## 1. Introduction

Maladaptive stress responses are common among college students. Between 27% [1] and 57% [2] of college students experience psychological distress, and 7.3% experience significant levels, including burnout [1]. These can affect their mental health, producing sleep disorders, depression, and anxiety [3], and impact academic performance and social functioning [4].

A lack of sleep is a stressor commonly reported by university students, conditioning greater exposure to stressful situations [5,6]. Insufficient sleep for five consecutive nights is sufficient to alter the evaluation of neutral and pleasant stimuli, leading to a more negative perception of them [7]. A lower perception of rest is associated with greater stress [8,9] and insomnia has been observed to mediate the relationship between stressors and emotional symptoms [10]. However, the pathways by which sleep increases stress vary among studies.

Resilience and self-control play an important role in the relationship between stress and sleep. Resilience is a process that denotes the ability to recover from adversity [11] and is considered the opposite of vulnerability to stress [12]. A consistent finding among university students was that higher levels of resilience were associated with lower levels of stress [13,14,15]. Sleep is also associated with resilience because it is considered the main cause of adequate brain function [16]. Resilience is associated with healthier indicators of sleep in both medical internship students [17] and nurses [18]; in contrast, people with insomnia have less resilience [19]. Therefore, a bidirectional relationship has been proposed between resilience and sleep. In adolescents, high levels of resilience predict better sleep quality; however, only sleep disturbance predicts worse resilience [20].

Self-control is a strategic behavioral mechanism involved in the adequate management of emotions and thoughts. This aids in adopting new behaviors or ceasing inappropriate ones [21]. Greater self-control is associated with less stress, which can reduce exposure to stressful situations [22,23,24] or improve coping responses [24,25]. Similarly, self-control has been implicated in physiological reactivity to stress, as adults with greater self-control have a better response (in blood pressure reactivity and recovery) to stressors [26]. Sleep also affects one’s self-control. Sleep-deprived individuals show altered activity in the prefrontal cortex, a brain area involved in self-control, inhibition, evaluation, and decision-making. A systemic review documented that sleep quality and duration are related to impulse control and how temptations are overcome [27]. Likewise, the capacity for self-control moderates the association between sleep quality the night before and emotional dissonance the following day when faced with negative emotional events [28]. Finally, self-control has also been linked to resilience, suggesting that it is a factor associated with fostering it [29].

Thus, the role of sleep on perceived stress is evident; however, certain variables that affect the stress response could also modify the effect of sleep. As resilience and self-control are two factors that are closely related to stress and sleep, we aimed to investigate the interaction or modification effects of self-control and resilience on the association between sleep and stress. We hypothesized that self-control and resilience would have a direct effect on stress and mediate the relationship between stress and sleep.

## 2. Materials and Methods

### 2.1. Setting and Participants

This cross-sectional study included 37 first-year students enrolled at a university in Mexico City. Participants were invited through convenience sampling. We used this sampling to ensure that the devices were used properly and returned to the college administrators. First, they were asked to respond to a survey and record their daily level of activity and sleep for six days using a wristband. Participation was voluntary, and all participants signed a letter of informed consent. Of the initial 37 students, one withdrew from the study, and four did not fully record their data and were eliminated. Thus, our study sample consisted of 32 participants, with a response rate of 86%. The study protocol was approved by the local research and ethics committee (FM-DI-065-2020).

### 2.2. Measurements

#### 2.2.1. Stress

We included ten items from the Mexican adaptation of the Perceived Stress Scale (PSS-10), a self-administered instrument that assesses the level of stress experienced during the last month. Each item is answered using a five-point Likert scale ranging from 0 = “never” to 4 = “very often”, with four inverse items, which added together provide the total stress rating. Higher scores indicate higher levels of perceived stress. Its psychometric properties, measured among Mexicans for the same age group as our participants (≤24 years), showed an α = 0.78, with two factors that explained 55% of the total variance [30]. The internal consistency of this scale in our sample yielded a Cronbach’s alpha score of 0.79.

#### 2.2.2. Resilience

Resilience was measured using the Brief Resilience Scale (BRS), a six-item scale that assesses the ability to recover from stress at a specific time [31]. Responses are coded using a Likert scale that ranges from 1 = “totally disagree” to 5 = “totally agree”, and includes three items that are inverted; the final score includes a summation of all items, and their average is calculated. Higher average scores indicated greater resilience. Normative data among Mexican college students yielded a Cronbach’s alpha score of 0.77 and a divergent validity with the Symptom Check List-90 (SCL–90) of r = −0.40 [32]. For our study sample, the reliability score was 0.81 using Cronbach’s alpha.

#### 2.2.3. Self-Control

The Spanish version of the Brief Self-Control Scale (BSCS) [33,34] was used to measure self-control. This self-administered scale includes 13 items, with responses ranging from 1 = “not at all agree” to 5 = “totally agree”. This version was tested on Latin American college students and showed an adequate fit with Tangney’s (2004) original model and internal consistency with an omega coefficient of 0.81 [34]. A sample of Mexican adolescents showed a Cronbach’s alpha score of 0.76 [35]. Our study sample using the BSCS yielded a Cronbach’s alpha coefficient of 0.74.

#### 2.2.4. Subjective Sleep Quality

The Pittsburgh Sleep Quality Index (PSQI) provides a global score of sleep quality and partial scores on seven different domains: subjective quality, latency, duration, habitual efficiency, disturbances, use of sleep medication, and daytime dysfunction. In our study, students were asked to respond to their experiences in these domains a month prior to PSQI administration. The individual scores for each of the seven domains ranged from 0 (no difficulty) to 3 (severe difficulty), and the global score ranged from 0 (no difficulty) to 21 (difficulties in all areas). A prior study among Mexican university students yielded a reliability score of α = 0.79 and was correlated with the Athens Insomnia Scale (r = 0.71; *p* < 0.0001) and the Epworth Sleepiness Scale (r = 0.47; *p* < 0.0001) [36]. In this study, the reliability score was α = 0.74.

#### 2.2.5. Sleep Duration

The Fitbit Charge 4TM (FBC) (Fitbit Inc., San Francisco, CA, USA) [37] is a wristband that allowed us to record activity through a three-axis accelerometer, an optical heart rate monitor, an altimeter, a vibration motor, and a relative SpO2 sensor for up to seven continuous days. Fitbit devices are widely used in research “https://www.fitabase.com/research-library/ (accessed on 2 May 2023)”. Students were asked to leave the wristband on their nondominant hand while sleeping to permit an analysis of their sleep patterns, displayed as the time spent in the rapid eye movement (REM), light sleep (LS), and deep sleep (DS) phases. Regarding the precision of this device, compared to polysomnography (PSG), it underestimates or overestimates the parameters of LS, DS, REM, and sleep efficiency (SE); however, it did not show differences with PSG in evaluating total sleep time (TST), waking after sleep onset (WASO), and sleep onset latency (SOL) [38]. Therefore, only WASO and TST were considered in this study. In addition, anthropometric data were collected when the study participants registered their use of the FitBit device on the manufacturer’s platform, enabling us to calculate their Body Mass Index (BMI).

### 2.3. Statistical Analysis

We used the IBM Statistical Package for the Social Sciences (SPSS), Version 23, New York, NY, USA [39], and EQS 6.3 software, Encino, CA, USA [40] to perform the statistical analyses. Stress, sleep quality (TST and WASO), resilience, and self-control are described using means and standard deviations. Correlations between variables were determined using Pearson’s test. Comparisons by sex were performed using the Mann–Whitney U test. For the main analysis, we tested whether the joint distribution of the variables included in the model met the normality criterion using the Mardia multivariate normality test. A path analysis technique was used to test the main study hypotheses, for which the assumptions of covariation between those included in the model were verified. The hypothesis tests were two-tailed, and the significance level was set at *p* < 0.05. Statistical power was calculated by considering the sample size (*n* = 32) and the expected association between sleep and perceived stress (r = 0.46) [28]. The achieved statistical power was 77.6%.

## 3. Results

Of the 32 students who had participated in this study, 21 (78%) were women and 11 (22%) were men, with a mean age of 18.47 (±0.84) years, single, and a mean BMI (by self-report) of 21.39 (±2.72). The average sleep time over the six days of recording by FBC was 5.9 (±0.9) hours. Twenty-three participants (72%) reported poor sleep quality (PSQI > 5). The characteristics of the study variables are presented in Table 1.

No differences were observed between women (W) and men (M) in the means (±standard deviation) of the main study variables. The mean differences were as follows: in the total PSQI (W = 7.6 ± 2.7; M = 7.2 ± 3.5; *p* = 0.735), TST (W = 356.1 ± 59.2; M = 365.0 ± 46.6; *p* = 0.736), PSS-10 score (W = 14.6 ± 5.7; M = 13.8 ± 4.4; *p* = 0.760), BSCS score (W = 46.4 ± 6.8; M = 43.0 ± 5.1; *p* = 0.267), and BRS score (W = 3.3 ± 0.8; M = 3.4 ± 0.4; *p* = 0.785).

The PSS-10 score (perceived stress) was significantly correlated with the BRS score (resilience) and BSCS score (self-control), as well as with the TST (sleep duration measured through the FBC) and PSQI total (sleep quality). Among the subjective aspects of sleep quality, higher latency, sleep disturbances, and daytime dysfunction were associated with a higher perception of stress (Table 1).

For the sleep quality model, we first assessed the joint distribution of the data followed by multivariate normality. The Mardia index confirmed that our data complied with this assumption (Mardia skewness = 17.06, *p* = 0.649; Mardia kurtosis = −0.59, *p* = 0.556).

The results indicated a good fit of the model; there were no significant differences between the empirical data and the proposed theoretical model (χ^2^ = 0.32; *p* = 0.572; CFI = 0.999; GFI = 0.995; AGFI = 0.949; and RMSEA = 0.001 [0.001–0.385]). The standardized coefficients are shown in Figure 1. We were able to observe that the presence of self-control and resilience reduces the size of the effect of the association between low sleep quality and perceived stress from 0.57 to 0.42. Similarly, lower quality of sleep among participants activates their self-control mechanisms (β = −0.43; *p* < 0.01), self-control activates resilience (β = 0.51; *p* < 0.01) and finally, resilience acts as a protective factor in the presence of stress (β = −0.56; *p* < 0.01) and low sleep quality.

To test the sleep duration model (TST), we first assessed the joint distribution of data assuming multivariate normality. The Mardia index confirmed that the data satisfied this assumption (Mardia skewness = 30.76, *p* = 0.058; Mardia kurtosis = 0.36, *p* = 0.721).

As a first approach, the modification indices suggested eliminating the covariation between self-control and stress because it was not significant; thus, we decided to remove it. In this way, the regression coefficient that showed a direct effect between TST and perceived stress was not significant, thus denoting a possible complete moderation of sleep duration in relation to perceived stress through self-control and resilience. Using the new model as a reference, we repeated the analysis. The results indicated a good fit of the model; that is, there were no significant differences between the empirical data and the proposed theoretical model (χ^2^ = 2.46; *p* = 0.292; CFI = 0.984; GFI = 0.962; IFI = 0.985; RMSEA = 0.010 [0.001–0.376]). The chi-square value suggests that there are no statistically significant differences between the theoretical and empirical models, while the incremental fit indices contribute to strengthening the working hypothesis. However, the RMSEA has values slightly higher than expected. The standardized coefficients are shown in Figure 2.

Our results showed that the presence of self-control and resilience enhanced the relationship between sleep duration and stress. Sleep duration activated self-control (β = 0.46; *p* < 0.01) and resilience (β = 0.09; *p* < 0.01). We noted a moderately positive association between resilience and self-control (r = 0.43; *p* < 0.01). Resilience acted as a protective factor against stress (β = −0.67; *p* < 0.01) in the face of an increase in sleep time. The results confirmed our main hypothesis, suggesting that the association between stress and sleep duration is completely moderated by resilience and self-control, such that when sleep duration increases, self-control and resilience are favored, contributing to a significant reduction in stress.

## 4. Discussion

Stress is highly prevalent among college students and can be increased by sleep disturbances. Exploring the mechanisms mediating this relationship could inform interventions aimed at reducing stress. The present study aimed to evaluate the mediating roles of self-control and resilience in sleep and perceived stress. We hypothesized that resilience and self-control could serve as buffers against low-quality sleep during stress responses. This hypothesis was confirmed via partial and total mediation models, in which self-control acted through resilience over perceived stress. To the best of our knowledge, this is the first study to evaluate resilience and self-control as mediating factors in the relationship between sleep and stress.

Duration and quality of sleep are correlated with stress, a finding widely reported in the literature that indicates how sleep buffers the effect of stressors on perceived distress and/or emotional well-being [8,9,10]. However, the mechanisms underlying the effects of sleep are poorly understood. Sleep was evaluated using different measures to obtain estimates from two perspectives. In the first approach, resilience and self-control mediated the relationship between sleep quality and perceived stress through partial mediation, in which sleep showed a direct effect on self-control and self-control influenced resilience. Resilience also directly affects stress. In the second model, self-control and resilience mediated the relationship between sleep duration and perceived stress. Systematic reviews have documented that low sleep quality and short sleep duration affect self-control [27]. Greater self-control in stressful situations favors the inhibition of impulses and the regulation of behavior, thoughts, and acts by decreasing exposure to stressful situations [21,41], fostering more adaptive coping behaviors [21,25,42], and improving sleep habits [43]. It is also associated with healthier physiological responses to stressors [26], facilitating recovery from stress, and strengthening resilience [29]. Similarly, the mediating role of self-control is consistent with findings among workers in Germany, in which greater self-control mediated the relationship between quality of sleep the night before and less discomfort, exhaustion, and emotional dissonance [28].

Moreover, greater resilience (less vulnerability to stress and greater recovery from environmental threats) affected perceived stress, confirming the association described in the literature. In other populations and among university students exposed to constant stressors, resilience is associated with reduced burnout [44] and distress [13,14,15].

Likewise, we observed that in simple correlations, sleep duration was related to resilience, but sleep quality was only weakly and not significantly related to resilience (which we attribute, mainly but not exclusively, to our small sample size). Studies are not consistent in this regard. A study on adolescents showed that only sleep disturbance predicted resilience [19]. In people undergoing shift changes, such as medical students [17] and nurses [18], resilience is consistently associated with various sleep characteristics; adults with insomnia also show low levels of resilience [20]. Therefore, it is plausible to put forth the hypothesis that the presence of sleep disturbance is strongly associated with resilience. Regarding self-control, both sleep duration and quality were significantly associated, confirming previous findings [27]. Sleep-related behaviors can also explain the association between self-control and sleep quality. One study showed that adults with a high capacity for self-control, in addition to better stress management, were more consistent in their use of sleep hygiene habits (i.e., sleep time regularity, avoiding sleep-disrupting food or activities, and maintaining a restful sleep environment); however, self-control was not associated with bedtime media use [43].

Finally, although college women reported more stress than men [45], our results did not show any sex differences. We believe that no differences were found because our measurements were taken at the beginning of the academic cycle, when the students had not yet experienced an increase in their stress levels. However, the lack of sex differences could also be a beta error due to the small sample size. Further research should consider both variables.

Study limitations. Our study had some limitations. This cross-sectional design did not allow us to establish causality. The small sample size and non-probabilistic selection limit the generalization of the results. However, the achieved statistical power (77.6%) was close to the expected minimum. This finding warrants further research on these variables. Therefore, we believe that our study provides an exploratory approximation of this topic. Another limitation is that women were overrepresented in our sample (in Mexico, health sciences are dominated by women at the undergraduate level). Furthermore, we did not have any control over menstrual cycles, which could have biased our results. Despite this, we believe our study findings could shed light on an under-researched topic in a middle-income country such as ours: a lack of sleep has deleterious effects on college students’ stress experiences.

## 5. Conclusions

The ability to be resilient seems to play a mediating role in the relationship between sleep and perceived stress; this ability can be favored, in turn, by self-control, which is directly influenced by sleep. Thus, a high capacity for self-control benefits resilience, and the effect of poor sleep on the stress response could be offset. We believe that more studies are needed to evaluate the mechanisms through which sleep can promote or decrease perceived stress, especially in vulnerable populations such as college students. This study provides an exploratory approximation of this topic.

## Figures and Tables

**Figure 1 ijerph-20-06560-f001:**
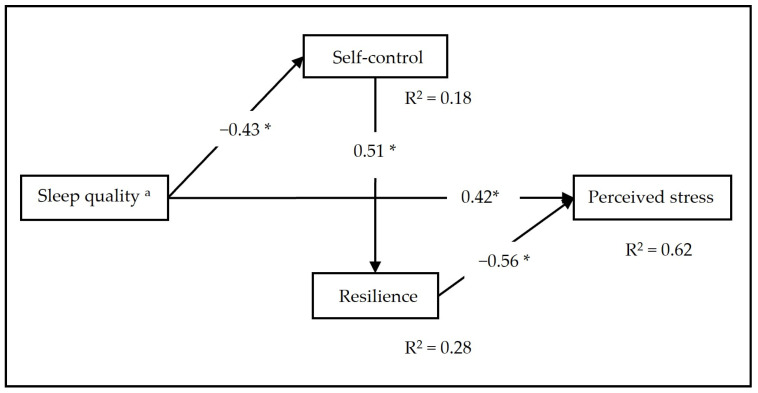
Partial mediation model of self-control and resilience and the relationship between sleep quality and perceived stress. Note: significance level of the standardized path coefficients (* *p* < 0.01) and explained variance (R^2^). ^a^ A higher score on the scale indicates worse sleep quality.

**Figure 2 ijerph-20-06560-f002:**
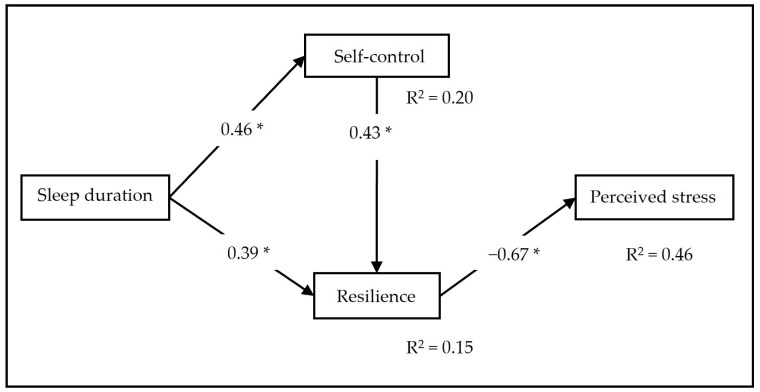
Mediation model of self-control and resilience and the relationship between sleep duration (TST) and perceived stress. Note: significance level of the standardized path coefficients (* *p* < 0.01) and explained variance (R^2^).

**Table 1 ijerph-20-06560-t001:** Descriptive values and correlations of the main study variables.

			Correlations		
	M	SD	PSS-10 Score (Perceived Stress)	BRS Score (Resilience	BSCS Score (Self-Control)
PSS-10 score (Perceived stress)	14.6	5.2	NA	NA	NA
BRS score (Resilience)	3.3	0.7	−0.63 **	NA	NA
BSCS score (Self-control)	45.0	6.4	−0.46 **	0.56 **	NA
Sleep (FBC, weekly average)					
WASO (min)	20.8	12.8	0.08	−0.08	−0.18
TST (min)	356.0	53.7	−0.35 *	0.36 *	0.36 **
PSQI total and subscales scores (sleep quality) ^a^:					
Overall sleep quality	7.8	2.8	0.57 **	−0.27	−0.43 *
Subjective sleep quality	1.3	0.4	0.26	−0.10	−0.29
Latency	1.4	0.9	0.59 **	−0.37 *	−0.48 **
Duration	1.6	0.9	0.01	0.08	0.06
Efficiency	0.8	1.0	0.07	0.05	−0.22
Sleep disturbances	1.0	0.5	0.43 *	0.03	−0.08
Use of sleep medication	0.3	0.5	0.28	−0.14	0.03
Daytime dysfunction	1.5	0.9	0.53 **	−0.34	−0.37 *

PSS-10 = Perceived Stress Scale; BRS = Brief Resilience Scale; BSCS = Brief Self-Control Scale; FBC = Fitbit Charge; NA= Not apply. ^a^ A higher score on the scale indicates worse/ poor sleep quality. * *p* < 0.05, ** *p* < 0.01.

## Data Availability

The datasets used and/or analyzed during the current study are available from the corresponding author on reasonable request.

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
