# Peer review of "Sleep and Perceived Stress: An Exploratory Mediation Analysis of the Role of Self-Control and Resilience among University Students"

_ijerph, 2023, doi:10.3390/ijerph20166560_

Round 1

Reviewer 1 Report

The authors attempted to determine if the association between sleep quality/duration and perceived stress was mediated by resilience and self-control using path analyses and found medication effects of these variables. While this is an interesting topic, the sample size used was small and recruitment strategy could lead to biased findings. Please see my detailed comments under different sections.

Introduction:

In general, the introduction is too long and distracting. Suggest shortening this giving only the essential background and current evidence succinctly and justifying the need for this study. The aim is also not clear due to the language used – not sure what “aimed to explore the role of these two factors in damping the effects of sleep on perceived stress” means. It suggests that the authors aimed to investigate any interaction or effect modification effects of self-control and resilience on the association between sleep and stress and is different from what was given as the aim under the methods. Furthermore, aims must be in the introduction/background section and not under methods.

Methods:

A sample size of 32 (or even 37) seems too small to draw mediating inferences. The authors must include what parameters were used in the sample size calculation and or what power did this sample had (again giving the parameters used) in detecting the desired associations.

It is also unclear why a convenience sampling was used in this study. Given the authors description, it seems that some form of random sampling was not impossible, and the authors have not justified the use of convenience sampling.

Was Brief Self-Control Scale and PSQI validated to be used in Mexico? This must be mentioned in the methods. 

The authors have reported demographic and anthropometric data in the results section but have not indicated under the methods how this information were collected.

Results:

Table 1 - The title is not descriptive enough. The scores of the tools that used to measure the constructs must be labelled as scored of those tools rather than as the construct themselves (as that is what the table shows) – e.g., PSS-10 score (Perceived stress), instead of Perceived stress (PSS-10). A digit (zero) must precede all decimal values.

Line #s 201-205 seem more methods than results. Suggest moving this section to the methods section.

Line #s 212-217 – What the authors were trying to explain in this section is not clear.

Give probability values for the calculated Mardia’s indices.

Discussion:

Line # 309 – The reference to meals and heavy activities is unclear. Suggest rewording that sentence.

While acknowledging that these findings could be biased and not generalisable, the authors could indicate potential practice and research implications of these findings.

Some English language editing is needed given the poor clarity in some sections. 

Author Response

RESPONSES TO REVIEWER # 1

Comments and Suggestions for Authors. The authors attempted to determine if the association between sleep quality/duration and perceived stress was mediated by resilience and self-control using path analyses and found mediation effects of these variables. While this is an interesting topic, the sample size used was small and recruitment strategy could lead to biased findings. Please see my detailed comments under different sections.

Comments

Response

Modification

Introduction:

In general, the introduction is too long and distracting. Suggest shortening this giving only the essential background and current evidence succinctly and justifying the need for this study.

Thank you for this suggestion, we will shorten the introduction to make it more succinct. Information we will remove or summarize in each of the paragraphs of the introduction.

This section was reduced from 879 to 524 words.

The aim is also not clear due to the language used – not sure what “aimed to explore the role of these two factors in damping the effects of sleep on perceived stress” means. It suggests that the authors aimed to investigate any interaction or effect modification effects of self-control and resilience on the association between sleep and stress and is different from what was given as the aim under the methods. Furthermore, aims must be in the introduction/background section and not under methods.

Thank you for this suggestion, we will ensure the language is consistent. The aim of the study will be highlighted in the Introduction and eliminated in the Methods section.

It now reads:

We aimed to investigate any interaction or modification effects of self-control and resilience on the association between sleep and stress.

Lines: 83-85

Methods:

A sample size of 32 (or even 37) seems too small to draw mediating inferences. The authors must include what parameters were used in the sample size calculation and or what power did this sample had (again giving the parameters used) in detecting the desired associations.

We agree with the reviewer in that ours was a small size. It was clarified in the study limitations. We will also expand on the potential selection biases. We will include a brief paragraph mentioning this is an exploratory study as well as include this information in the title.

It now reads:

Sleep and perceived stress: A exploratory mediation analysis of the role of self-control and resilience among university students

Lines: 2-4

It now reads:

Thus, we believe ours is an exploratory approximation to this topic.

Lines: 294-295

It is also unclear why a convenience sampling was used in this study. Given the authors description, it seems that some form of random sampling was not impossible, and the authors have not justified the use of convenience sampling.

We used convenience sampling because we needed to make sure the devices were used properly, and data was also collected properly. We agree with this suggestion and will include in the text, an additional explanation on why we were unable to carry out random sampling.

It now reads:

We used convenience sampling because we needed to ensure the devices were used properly and returned to the college administrators.

Lines: 91-93

Was Brief Self-Control Scale and PSQI validated to be used in Mexico? This must be mentioned in the methods.

Thank you for pointing this out. The surveys we included were validated in Mexico and this will be further clarified and referenced in the Methods section.

It now reads:

This version was tested in Latin American college students and showed an adequate fit with Tangney’s (2004) original model and an internal consistency with an Omega coefficient of 0.81 [34]. Among a sample of Mexican adolescents, it showed a Cronbach’s Alpha score of 0.76 [35].

Lines: 123-126

The authors have reported demographic and anthropometric data in the results section but have not indicated under the methods how this information was collected.

We apologize for not discussing this in more detail. Anthropometric data for this study was collected when study participants registered their use of the FitBit device on to the manufacturer’s platform. We thought we explained this when we mentioned the information collected was by self-report but wewill ensure this is clearly described in the Methods section.

It now reads:

 In addition, anthropometric data was collected when study participants registered their use of the FitBit device on to the manufacturer’s platform, and this allowed us to calculate their Body Mass Index (BMI).

Lines: 151-153

Results:

Table 1 - The title is not descriptive enough. The scores of the tools that used to measure the constructs must be labelled as scored of those tools rather than as the construct themselves (as that is what the table shows) – e.g., PSS-10 score (Perceived stress), instead of Perceived stress (PSS-10).

A digit (zero) must precede all decimal values.

We thank you for suggesting we make the title of Table 1 more readable.  We will include the scores of the tables we used and will include a zero before all decimals.

It now reads:

Descriptive values and correlations of the main study variables

Line: 174

Now a digit (zero) precedes all decimal values.

See: Table 1

Line #s 201-205 seem more methods than results. Suggest moving this section to the methods section.

We agree with the reviewer and will make the suggested change

The information was eliminated because it was already indicated in the methods section.

Line #s 212-217 – What the authors were trying to explain in this section is not clear.

We apologize for not making the text readable. We will edit as suggested.

Se eliminó el texto de las líneas 214-217, por considerarlo confuso e innecesario.

Give probability values for the calculated Mardia’s indices.

We agree with the reviewer and will include Mardia’s indices

It now reads:

The Mardia index confirmed our data complied with this assumption (Mardia skewness= 17.06, p= 0.649; Mardia kurtosis= -0.59, p= 0.556).

The Mardia index confirmed the data met this assumption (Mardia skewness= 30.76, p= 0.058; Mardia kurtosis= 0.36, p= 0.721). 

Lines: 189-191, 206-207

Discussion:

Line # 309 – The reference to meals and heavy activities is unclear. Suggest rewording that sentence.

We agree with the reviewer and will make the suggested change in line 309.

It now reads:

A study showed that adults with a high capacity for self-control, in addition to manag-ing stress better, were more consistent in the use of sleep hygiene habits (ie., sleep time regularity, avoiding sleep-disrupting food or activities, and keeping a restful sleep envi-ronment); however, self-control was not associated with bedtime media use [43].

Lines: 280-284

While acknowledging that these findings could be biased and not generalizable, the authors could indicate potential practice and research implications of these findings.

As mentioned above, we will further clarify the research implications of our results and its potential for generalizability.

It now reads:

However, the statistical power we achieved (77.6%) is near the expected minimum. This warrants further research with these variables in other samples. Thus, we believe ours is an exploratory approximation to this topic. Another study limitation is that women were overrepresented in our sample (in Mexico, at an undergraduate level, health sciences are dominated by women). Furthermore, we did not have any control of their menstrual cy-cles which could have biased our results. In spite of this, we believe our study findings could shed light on an under researched topic in a middle-income country such as ours; lack of sleep has deleterious effects on college students’ quality of life.

Lines: 292-300

Comments on the Quality of English Language:

Some English language editing is needed given the poor clarity in some sections.

We apologize and will review the manuscript more thoroughly as to facilitate reading.

Reviewer 2 Report

Tafoya et. al., report the role of self-control and resilience in damping the effects of sleep on perceived stress among university students. Based upon mediation analysis, authors show that self-control benefits resilience, and both acting like buffers can offset the effect of poor sleep on the stress response. Authors also observed that sleep duration but not the quality was better related to resilience. Their findings regarding self-control validated previous reports of a significant association between both sleep duration and quality.

As authors have mentioned, a major concern is generalization of the results from small sample size and its non-probabilistic selection method. Can authors comment about the statistical power of their study?

Apart from that, it is an overall interesting and well-presented manuscript. Rationale for such a study is highlighted in the introduction, and methods, results and discussion are adequately detailed with recent and relevant references.

I have few other minor concerns-

1. Considering a high proportion of female participants which could skew the findings, authors should include a brief discussion with few relevant studies which can highlight the presence or absence of any gender differences for stress perception, resilience and self-control. Any particular reason for a greater number of female participants? Participant’s field of study (major subject) can be included in the demographics.

2. Authors need to clarify whether the study period of 6 days did or did not coincide with menstrual period of female participants as it could have significant impact of overall stress physiology.

3. Terms bracelet and watch-type can be simply replaced with wrist-band.

Author Response

RESPONSES TO REVIEWER # 2

Comments and Suggestions for Authors. Tafoya et. al., report the role of self-control and resilience in damping the effects of sleep on perceived stress among university students. Based upon mediation analysis, authors show that self-control benefits resilience, and both acting like buffers can offset the effect of poor sleep on the stress response. Authors also observed that sleep duration but not the quality was better related to resilience. Their findings regarding self-control validated previous reports of a significant association between both sleep duration and quality. As authors have mentioned, a major concern is generalization of the results from small sample size and its non-probabilistic selection method. Can authors comment about the statistical power of their study? Apart from that, it is an overall interesting and well-presented manuscript. Rationale for such a study is highlighted in the introduction, and methods, results and discussion are adequately detailed with recent and relevant references.

Comments

Response

Modification

As authors have mentioned, a major concern is generalization of the results from small sample size and its non-probabilistic selection method. Can authors comment about the statistical power of their study?

We agree with the reviewer´s comment, and this was also pointed out by the other reviewer. We will include a more detailed explanation in the text that considers how we used convenience and not probabilistic sampling.

Regarding statistical power, this was included,

Lines: 164-166, 292-293

1a. Considering a high proportion of female participants which could skew the findings, authors should include a brief discussion with few relevant studies which can highlight the presence or absence of any gender differences for stress perception, resilience and self-control.

We thank the reviewer for pointing out this very interesting topic. In Mexico, at an undergraduate level, health sciences are mostly dominated by women. This is the reason behind the overrepresentation in our study. We agree that women respond differently to stress and will include additional references and point this out in the study limitations.

It now reads:

No differences were observed between woman (W) and man (M) in the means (± standard deviation) of the main study variables. In the total PSQI (W= 7.6±2.7, M= 7.2±3.5; p=0.735), in the TST (W= 356.1±59.2, M= 365.0±46.6; p=0.736), in the PSS-10 score ( W= 14.6±5.7, M= 13.8±4.4; p=0.760), BSCS score (W= 46.4±6.8, M= 43.0±5.1; p=0.267), and in the BRS score (W= 3.3±0.8, M = 3.4±0.4; p=0.785).

Finally, even though college women report more stress than men [45] our results did not show gender differences. We believe that no differences were found because our measurements were taken at the beginning of the academic cycle when the students had not yet experienced an increase in their stress levels. However, further research should take into consideration both variables.

Lines: 178-182, 285-289

1b. Any particular reason for a greater number of female participants? Participant’s field of study (major subject) can be included in the demographics.

It now reads:

Another study limitation is that women were overrepresented in our sample (in Mexico, at an undergraduate level, health sciences are dominated by women).

Lines: 295-296

2. Authors need to clarify whether the study period of 6 days did or did not coincide with menstrual period of female participants as it could have significant impact of overall stress physiology.

We thank the reviewer for pointing out this very important question. We will ensure the Fitbit device did in fact register this data.

We did not collect this information, and this will be included as an additional limitation and one that will be considered for further studies.

It now reads:

Furthermore, we did not have any control of their menstrual cycles which could have biased our results. Lines: 297-298

3. Terms bracelet and watch-type can be simply replaced with wrist-band.

We agree and thank the reviewer´s for this suggestion. The term was replaced in the text.

See lines: 32, 95, 140, 144

Round 2

Reviewer 1 Report

This manuscript has significantly improved from the previous version. A few other comments for further improvement are given below.

Results

In Figure 1, the direct effect of sleep quality on stress seems not significant. It is not clear why the authors have kept this but removed the link between TST and stress in Figure 2.

Discussion

Line #s 270-271: It is not clear what the authors meant by “An explanation could be found in the participant’s dream characteristics.” This seems irrelevant ant out of place. The section from line #s 271-285 is also confusing and its relevance if not clear, referring to multiple outcomes that are different from the constructs assessed in this study. Suggest summarising any available and relevant evidence with a sentence or two that clearly relate them to the constructs investigated in this study.

 Line #s 286-289: The lack of gender differences could also be a beta error due to small numbers. Suggest incorporating that fact.

Line #s 300-301: This last sentence if also irrelevant – again, quality of life was not investigated in this study.

Unfortunately, the language issues still persist including grammatical errors starting from the title. Some sections are difficult to understand because of these language issues. Some sentences are confusing due to seemingly inappropriate use of common English words. I recommend getting help of an English language expert/native speaker to remedy these issues. 

Author Response

The authors appreciate the observations made by the reviewer. They greatly improved the manuscript.

Results

In Figure 1, the direct effect of sleep quality on stress seems not significant. It is not clear why the authors have kept this but removed the link between TST and stress in Figure 2.

Figure 1 was corrected to be consistent with Figure 2

Discussion

Line #s 270-271: It is not clear what the authors meant by “An explanation could be found in the participant’s dream characteristics.” This seems irrelevant ant out of place.

We agree with the observation. This line was removed.

The section from line #s 271-285 is also confusing and its relevance if not clear, referring to multiple outcomes that are different from the constructs assessed in this study. Suggest summarising any available and relevant evidence with a sentence or two that clearly relate them to the constructs investigated in this study.

The information was summarized, and we hope we clarified the idea we were trying to  highlight.

Line #s 286-289: The lack of gender differences could also be a beta error due to small numbers. Suggest incorporating that fact.

The text was added

Line #s 300-301: This last sentence if also irrelevant – again, quality of life was not investigated in this study.

The line was corrected

Unfortunately, the language issues still persist including grammatical errors starting from the title. Some sections are difficult to understand because of these language issues. Some sentences are confusing due to seemingly inappropriate use of common English words. I recommend getting help of an English language expert/native speaker to remedy these issues.

The text was sent to English language editing
